# Responses of Rice Growth to Day and Night Temperature and Relative Air Humidity—Dry Matter, Leaf Area, and Partitioning

**DOI:** 10.3390/plants8110521

**Published:** 2019-11-18

**Authors:** Sabine Stuerz, Folkard Asch

**Affiliations:** Institute of Agricultural Sciences in the Tropics (Hans-Ruthenberg-Institute), University of Hohenheim, 70593 Stuttgart, Germany; fa@uni-hohenheim.de

**Keywords:** growth chamber, leaf mass fraction, *Oryza sativa*, root mass fraction, specific leaf area, stem mass fraction

## Abstract

Asymmetric changes of day and night temperature have already been observed because of Climate Change. However, knowledge on environmental conditions either during day or night serving as trigger for growth processes is scarce. In this study, one rice (*Oryza sativa*) variety (IR64) was examined to assess the impact of varying temperatures and relative air humidities during day and night periods on biomass, leaf area, and dry matter partitioning between organs. Three different day and night temperature (30/20 °C, 25/25 °C, 20/30 °C) and relative air humidity (40/90%, 65/65%, 90/40%) regimes were established. The effect of relative air humidity on both plant dry matter and leaf area was larger than the effect of temperature, in particular low humidity had a strong negative impact during the night. With high day temperature, the shoot mass fraction increased, whereas the root mass fraction decreased. Specific leaf area increased at high night temperatures and led, along with the high leaf mass fraction at high night humidities, to higher growth rates. The results emphasize the importance of considering relative air humidity when focusing on plant responses to temperature, and strongly suggest that under asymmetric day and night temperature increases in the future, biomass partitioning rather than biomass itself will be affected.

## 1. Introduction

With climate change, temperature increases are not only expected in the coming decades, but have already been observed [1]. Of particular concern is that average temperature increases could compromise the world’s rice production. Masutomi et al. [2] predicted rice yield losses for most of the rice growing regions in Asia in the near future, and Lobell et al. [3] identified rice in Southeast Asia as one of the most important crops in need of climate change adaptation investment. Whereas the increase of average temperature and its negative effects on rice yields are widely accepted, there is no consensus about the effects of changes to the temperature range, the difference between day and night temperature on a daily basis. Despite high uncertainty [1], a decrease in the daily temperature range is expected in the world’s rice growing regions [4] as daily minimum temperatures increase more rapidly than daily maximum temperatures [5]. Negative yield responses of rice to a narrowing daily temperature range due to an increase in minimum temperature have been observed in field experiments in the Philippines [6]. However, Lobell [4] projected impacts of a narrowing daily temperature range on rice yield as positive, even if relatively small.

As shown by the contradictory results of the two studies, the role of night temperature per se is not well understood [7]. Experimental approaches aiming at exploring the effect of high night temperature on plant growth face the difficulty that only increasing night temperature means an increase in daily mean temperature, which also has an effect on plant growth and development, making it difficult to discriminate between both effects. To exclude the effects of a higher mean temperature, varying daily temperature ranges around the same mean temperature can be used to study plant responses. In such an experimental setup, the possibility of including high night temperatures is limited, since this would require a high mean temperature, which would lead to heat stress when using a large daily temperature range. To avoid heat stress effects during the day, but to test plant responses to a wide range of night temperatures, day and night temperature can be inverted, an approach that has been used in this study.

Another constraint of studies on high night temperature under natural conditions is the physical interaction of temperature and relative air humidity. Since minimum night temperature is usually around the dew point, a smaller daily temperature range is directly correlated with higher relative air humidity during the day. However, changes in relative air humidity have been associated to changes in photosynthetic rates [8], dry matter and leaf area [9], and spikelet sterility [10] and thus, the potential effect of relative air humidity on the rice plant should not be disregarded in studies on temperature effects.

In a field study, high night temperature led to biomass reduction in one of the two rice genotypes [11], whereas Peraudeau et al. [7] reported no changes in total dry matter, increased leaf area for indica varieties in one out of the three experiments, but consistently lower SLA (specific leaf area) under elevated night temperature. In a meta-analysis, Jing et al. [12] found that plant leaf growth was increased by high night temperature as well as leaf area ratio (LAR) and specific leaf area (SLA), while the effect on organ weight and total dry matter was less clear, and varied between plant functional groups. As a result, it was concluded that complexities and challenges remain when seeking general patterns of plant growth in response to high night temperature. Little information exists on the effect of temperature on partitioning in rice plants. In a climate chamber experiment, high night temperature during reproductive growth led to increased dry weight per hill, because of a higher stem weight, whereas leaf and root dry weight was not affected by temperature [13]. Increased daily temperature in open-top chambers in the field led to higher leaf area, but the effects on partitioning of dry weight between organs was not consistent [14]. In summary, there seems to be a larger effect of temperature on leaf area than on total dry matter, but the impact on dry matter partitioning between organs remains unclear. In all cited studies, temperature was raised during the night, in one case during day and night, with the result of a higher daily mean temperature compared to the control treatment. Hence, effects of day, night, and mean temperature cannot be distinguished.

As plants must achieve a balance between carbon assimilation (occurring during the day) and storage and growth (occurring during both day and night) [15], there are likely differential effects of day and night temperature on carbon allocation. For example, high night temperature could enhance growth processes during the night, which in turn could induce the buildup of more photosynthetic tissue to meet the increased demand. Furthermore, high day temperature results in a higher water demand, which in turn could favor root growth. However, these processes have hardly been studied yet, even though greater insight into them could help predict the consequences of Climate Change for rice production. Therefore, the objective of this study is to disentangle effects of day, night, and mean temperature on dry matter and its partitioning between organs and leaf area, while also taking into account the effects of relative air humidity by using inverted day/night temperatures and air humidities.

## 2. Results

Rice plants were grown at three different temperature regimes with either “natural” (30 °C/20 °C; Tnat), constant (25/25 °C; Tcon), or inverted (20/30 °C; Tinv) day/night temperature in combination with three different relative air humidity (RH) regimes with either “natural” (40/90%; RHnat), constant (65/65%; RHcon), or inverted (90/40%; RHinv) day/night RH. Two additional treatments with either constantly low (20/20 °C; Tcon-l) or high (30/30 °C; Tcon-h) temperature, both at RHcon, were established. After two weeks of different day/night temperature and RH treatment with the same mean temperature and RH, total plant dry matter varied between 6.0 g for Tnat/RHcon and 10.4 g for Tnat/RHnat (Table 1).

Temperature had no statistically significant effect (Table 2), but RHnat led to a significantly higher total dry matter per plant with an average 9.7 g versus 7.8 g and 8.1 g under RHcon and RHinv, respectively. At Tnat, RHnat led to a statistically significant higher total dry matter compared to RHcon, whereas RH had no statistically significant impact in the other temperature treatments.

Much larger variation was observed in leaf area, which ranged from 333 cm^2^ under Tcon/RHinv to 890 cm^2^ under Tinv/RHcon. Temperature only had a significant effect under RHcon, with the highest leaf area at Tinv (890 cm^2^), and the lowest at Tnat (341 cm^2^). RH had a larger effect on leaf area than temperature, with RHinv leading to a significantly lower leaf area of 356 cm^2^ on average in comparison to 621 cm^2^ and 596 cm^2^ under RHnat and RHcon, respectively. Under constant temperature and relative humidity, both total dry matter and leaf area were significantly lower at 20 °C (2.9 g; 162 cm^2^) than at 25 °C (7.9 g; 558 cm^2^) and 30 °C (9.2 g; 439 cm^2^) (Table 3).

Leaf mass fraction (LMF), the leaf dry weight per plant dry weight, did not respond to temperature, but RH had a statistically significant effect (Figure 1).

With an average of 0.20 g g^−1^, the LMF was significantly lower under RHinv than under RHnat with 0.24 g g^−1^ and RHcon with 0.27 g g^−1^ (Figure 1). Under the constant temperature regimes, LMF was significantly lower at 30 °C (0.19 g g^−1^), than at 20 °C (0.23 g g^−1^), and 25 °C (0.25 g g^−1^) (Figure 1, Insert 1). Temperature had a significant effect on the stem mass fraction (SMF) (Figure 1, Insert 2), which is the fraction of stem dry weight per plant dry weight. Under Tinv, it was significantly lower (0.37 g g^−1^, on average) than under Tnat and Tcon, which both resulted in a SMF of 0.41 g g^−1^. Also RH had an effect on SMF, which was significantly lower (0.38 g g^−1^) under RHcon than under RHnat (0.40 g g^−1^) and RHinv (0.41 g g^−1^). Constant temperature between 20 and 30 °C did not have any significant effect on SMF.

The root mass fraction (RMF), the root dry weight per plant dry weight, varied between 0.31 g g^−1^ for Tnat/RHnat and Tnat/RHcon and 0.38 g g^−1^ for Tinv/RHnat and Tinv/RHinv and was strongly influenced by temperature. Under Tnat, RMF was on average 0.31 g g^−1^ and significantly lower than under Tcon and Tinv with 0.34 g g^−1^ and 0.37 g g^−1^, respectively. RH as well as different constant temperatures between 20 and 30 °C had no effect on RMF. The faction of dead leaves (DLMF) varied between 0.00 g g^−1^ for Tinv/RHcon and 0.06 g g^−1^ for Tnat/RHcon and Tnat/RHinv. Nevertheless, there were no significant differences because of large variation within treatments.

The SLA varied largely and ranged from 19.7 m^2^ kg^−1^ (Tnat/RHinv) to 32.5 m^2^ kg^−1^ (Tinv/RHcon) (Figure 2).

Both temperature and RH had large effects on SLA. On average, Tnat led to the lowest SLA with 22.6 m^2^ kg^−1^ followed by Tcon with 25.9 m^2^ kg^−1^ and Tinv with 28.8 m^2^ kg^−1^. Among the RH treatments, SLA was significantly lower under RHinv with 22.8 m^2^ kg^−1^ than under RHnat and RHcon with 26.6 and 28.0 m^2^ kg^−1^, respectively. Further, a significant interaction effect between temperature and RH was found. Under Tnat, RHnat resulted in a significantly higher SLA compared to RHinv, whereas under Tcon and Tinv, RHcon resulted in a significantly higher SLA compared to RHinv. At RHnat, temperature had no significant effect on SLA, while at RHcon, Tinv and Tcon resulted in a significantly higher SLA than Tnat and at RHinv, Tinv resulted in a significantly higher SLA than Tnat. Among the constant temperature treatments, no significant difference was found in regard to SLA.

Regressing SLA and plant organ fractions versus day and night temperature and RH of all treatments, including the different constant temperatures, resulted in a significant positive correlation between SLA and night temperature (*r* = 0.65; *p* = 0.030) (Table 4).

The stem mass fraction was positively correlated with day temperature (*r* = 0.76; *p* = 0.006) and the root mass fraction was negatively correlated with day temperature (*r* = −0.84; *p* = 0.001) and positively with night temperature (*r* = 0.63; *p* = 0.037). Leaf and dead leaf mass fractions were not correlated with any of the parameters.

## 3. Discussion

### 3.1. T and RH Effects on Plant Dry Matter and Leaf Area

Under constant RH, inverted day and night temperatures led to increased leaf area. Provided that plant growth is driven by photosynthetic carbon fixation during the day [16], higher growth rates could be expected under higher day temperature, at least in the temperature range of our experiment, since maximal assimilation rates for rice were found in the range of 30–35 °C irrespective of the growth temperature [17]. Furthermore, higher night temperature increases respiration, which consumes a large fraction of daily available assimilates and thus limits biomass production [18]. Contrastingly, stomatal conductance [19] and leaf area development [20] of rice were found to be positively correlated with night temperature in a semi-arid environment. At this point in time, we cannot explain the larger leaf area under inverted day and night temperature and further research is needed to elucidate the link between diurnal temperature pattern, photosynthesis, and growth response.

Different day and night temperatures alone had no significant effect on total plant dry matter, but significantly affected the partitioning between plant organs and leaf area. While Peraudeau et al. [7] found increased leaf area and constant total plant dry weight under high night temperature, we observed an increase in leaf area under high night temperature only under constant RH. Under natural conditions, RH at night is usually close to 100% and substantially lower during the day. In temperature-controlled/heated greenhouses or growth chambers, diurnal RH often fluctuates less, since even though the absolute amount of water in the air remains constant, RH decreases with increasing temperature because of heating. Therefore, our results indicate that temperature responses observed in temperature-controlled environments may not be applicable to field-conditions.

However, RH not only influenced the plants’ growth response to temperature, it also had a strong direct effect on dry matter and leaf area, which were both highest at low day/high night RH. In an experiment conducted by Hirai et al. [21], high humidity during the light period in combination with low humidity during the dark resulted in higher total dry matter than in other combinations of high and low, day and night RH, but they found a positive effect of high RH on plant growth in general. Equally, Kuwagata et al. [9] described a positive effect of high RH on dry matter production of rice and found that low RH induced the upregulation of many *PIP* and *TIP* aquaporin genes. Since the mRNA levels of root aquaporins reach a maximum 2 h after the onset of the light period [22], upregulation of aquaporins may only occur in response to dry days and not to dry nights. If there is a positive effect of low RH during the day or a negative effect of low RH during the night, it cannot be clearly answered from our dataset. RH only directly affects the plant’s transpiration, which is only active during the day, as stomatal opening is induced by light. Therefore, a negative effect of low RH during the night does not seem likely. However, as the lowest leaf mass fraction, the highest fraction of dead leaves, and the lowest SLA were observed under low RH at night (Figure 2), we hypothesize that low RH during the night, rather than high RH during the day, resulted in strongly reduced biomass and leaf area under the inverted RH regime.

### 3.2. T and RH Effects on Partitioning

Decreases in LMF, leaf abscission, and reduced SLA were found under RHinv. These are adaptation strategies to reduce leaf area under water deficit. Thus, under the experimental setting these are likely to occur only in low RH environments. Furthermore, dry weight of dead leaves and SLA were strongly negatively correlated (Figure 3), indicating that both phenomena have a common cause.

Even though plants grown under low RH showed morphological adaptation to water deficit, it was not because of the water stress, as transpiration happens during the day. SLA is instead likely regulated via a plant parameter directly related to RH such as leaf water potential, triggered by low RH at night.

SLA was not only highly affected by RH. Across all RH treatments, SLA was lowest under “natural” and highest under inverted temperature. Although a significant positive correlation was found between SLA and night temperature, we hypothesize that the relationship between day and night temperature rather than night temperature itself controls SLA. This was supported by the lack of difference in SLA in treatments with constant day and night temperatures. Published data on SLA responses to temperature are contradictory. For example, Kuwagata et al. [9] reported that low root zone temperatures decreased SLA, whereas Peraudeau et al. [7] showed a decrease in SLA under high night temperatures. In contrast to the results presented here, Sunoj et al. [23] reported an increase in SLA with increasing temperature amplitude and higher day time temperatures. Tardieu et al. [24] argued that SLA decreases when leaf expansion is more affected by environmental conditions than photosynthesis. Because of the decrease in SLA during warm and/or humid days, conditions that should be beneficial to photosynthesis and leaf expansion, any reduction in SLA must be due to low expansion rates during the night, triggered by low temperature and low RH.

The RMF increased with higher night temperature and decreased with higher day temperature. The fraction of roots was not negatively correlated with total plant dry matter, a relationship described as the ontogenetic drift, when larger plants invest a larger fraction of their biomass in support structures [25]. This would support the hypothesis that changes in the RMF are actually temperature driven. Since RH had no effect on the RMF, and the highest RMF was found at low day temperature, water demand of the shoot can be excluded as the direct driving force for root growth in our experiment. As a low RMF did not correspond with a reduction in growth, it suggests the presence of a compensation mechanism, such as increased hydraulic conductivity per unit root weight. Kuwagata et al. [9] found that low temperature reduced the quantity of roots and argued that the reduced surface area of the roots was compensated by an increased water uptake per unit root volume facilitated by a higher expression levels of root-specific aquaporin genes. Therefore, it seems more likely, that RMF increased with higher night temperature than it decreased with higher day temperature. However, a final conclusion cannot be drawn from the dataset. The SMF increased with higher day temperature. In contrast to the RMF, the SMF not only responded to a diurnal change in temperatures, but also to the constant temperature treatments. We hypothesize that the higher SMF under the “natural” diurnal temperature regime is a result of warm days and not from cool nights, because of the increase in the SMF at higher temperatures when the day and night temperatures are constant. Cheng et al. [13] found an increased stem weight under high night temperature, while root and leaf dry weight were unaffected. In maize, an increased total dry matter was reported at a larger daily temperature range [23], but it is unclear if this resulted from higher day temperature or from reduced respiration at lower night temperature.

Effects of the daily temperature and RH pattern on biomass partitioning and SLA were clearly shown. Results are summarized in Figure 4.

In order to link the observed parameters with growth rates, regression analyses were performed (Table 5).

None of the parameters was significantly correlated with dry matter accumulation. However, the increase in leaf area was significantly positively correlated with SLA and LMF, and negatively correlated with DLMF. In wheat, higher SLA equated with faster leaf area production [26]. Since both, high SLA and high LMF increase the photosynthetic area of a plant, light capture, and ultimately carbon gains increase.

### 3.3. Limitations of the Study

The main objective of this study was to evaluate the effect of high night temperature on biomass partitioning between plant organs, an issue of particular interest for an important crop such as rice as night temperature increases are predicted around the globe. Nevertheless, the most important question, how the partitioning into reproductive organs will be affected, was not addressed. Across different plant species, high night temperature was related to a reduction in biomass allocation to reproduction organs [12]. For rice, a reduction in grain growth rate during grain filling was found using inverted day and night temperatures [27]. However, under high night temperature, increased biomass accumulation, because of higher growth rates during the vegetative stage could overcompensate an impaired reserve partitioning into grains. Further research is needed to address this question.

Furthermore, in this study, different day and night temperatures were only tested around one mean temperature. The temperature range 20–30 °C in this experiment was chosen because it represents a large temperature range and does not lead to cold or heat stress effects. Under different mean temperatures, temperature stresses would most probably lead to different results. Also only one variety was used in this experiment. IR64 was ranked as medium tolerant to high night temperature [28], and the test of different varieties differing in their temperature sensitivity would definitely improve our understanding of crop responses to increasing night temperatures.

Finally, temperature fluctuations in the rooting zone are by far not as pronounced in the field, as in our experimental set-up, and therefore responses to temperature will be less distinct, at least for below-ground parts of the plant. On the other hand, temperature and RH probably affect the plant’s water status to a larger degree when grown in soil, especially under water-limited conditions, and this will be relevant for the entire plant.

## 4. Materials and Methods

### 4.1. Plant Cultivation

The experiment was conducted in plant growth chambers at the Institute of Agricultural Sciences in the Tropics of the University of Hohenheim, Germany. In total, eleven sets of plants were used. For each, seeds of one variety (IR64) were germinated in petri dishes on filter paper for one week. Individual seedlings were transferred into pots containing 1 l of half strength nutrient solution in the composition proposed by [29]. After another week, half strength nutrient solution was replaced by full strength nutrient solution and, from then onward, full strength nutrient solution was exchanged every week. Plants were grown in a growth chamber (Percival Intellus Environmental Controller—EA-75HIL) at 28/22 °C day/night temperature, a mean temperature of 25 °C and a mean relative air humidity (RH) of 75% for the first five weeks. Afterward, plants were transferred to another growth chamber (Percival Intellus Ultra Controller—E-75L1), where each set of plants was cultivated under different environmental conditions (Table 6) for a duration of two weeks: a day/night temperature regime of 30/20 °C, considered similar to natural conditions (Tnat); 25/25 °C, a constant temperature regime (Tcon); and 20/30 °C, an inverted temperature regime (Tinv). The RH was set to a day/night regime of 40/90%, considered similar to natural conditions (RHnat); 65/65%, a constant RH (RHcon); and 90/40%, an inverted RH regime (RHinv). The combination of three temperature and three RH settings resulted in nine treatments, each experiencing a daily mean temperature of 25 °C and 65% RH. Additionally two temperature settings were established with a constantly lower temperature of 20 °C (Tcon-l) and a constantly higher temperature of 30 °C (Tcon-h) and constant RH of 65% in order to distinguish effects of day/night and mean temperature. Temperature and RH were recorded with TinyTag TGP-4500 Dual Channel data loggers (Gemini Co., Chichester, UK) in both chambers.

### 4.2. Measurements

For each set of plants, destructive samplings were conducted on three plants at the onset of treatments (0 DOT) and at 14 DOT. Plants were individually separated into green leaf blades (hereinafter referred to as leaves), leaf sheaths (hereinafter referred to as stems), roots, and dead leaves. Leaves were considered as dead, when the leaf blade was either completely yellow or by at least 50% dry and dead. Leaf area of all green leaves was measured with a LI-COR leaf area meter (LI-3000C) in combination with a belt conveyer accessory (LI-3050C). All plant material was dried at 70 °C for at least 48 h and weighed. Specific leaf area (SLA) was calculated as green leaf area per kg of leaf dry matter of the whole plant [m^2^ kg^−1^].

### 4.3. Data Analysis

Data analysis was carried out with STATISTICA 13 [30] using analysis of variance (ANOVA) followed by a Tukey HSD post-hoc test to analyze differences between treatments for total dry matter, leaf area, mass fractions, and SLA. For the comparisons of temperature and humidity effects at the same mean temperature and humidity, a factorial ANOVA was used, while for the comparison of temperature effects in the treatments of constant temperature a one-way ANOVA was used. All levels of significance were set to *p* < 0.05.

## 5. Conclusions

High night temperature resulted in higher leaf area, but only under constant relative air humidity, whereas under low air humidity during the day and high air humidity during the night, high night temperature had no effect on biomass or leaf area. Nevertheless, diurnal temperatures highly affected the partitioning between plant organs. The fraction of stems increased with higher day temperature, whereas the fraction of roots increased with higher night temperature. The leaf mass fraction was only affected by the day and night pattern of relative air humidity, but SLA strongly responded to both, temperature and relative humidity. The higher leaf mass fraction during humid nights, and higher SLA during warm nights were associated with higher leaf growth rates. Therefore, we argue that rice plants might benefit from an asymmetric day/night temperature increase, at least during vegetative growing phases.

## Figures and Tables

**Figure 1 plants-08-00521-f001:**
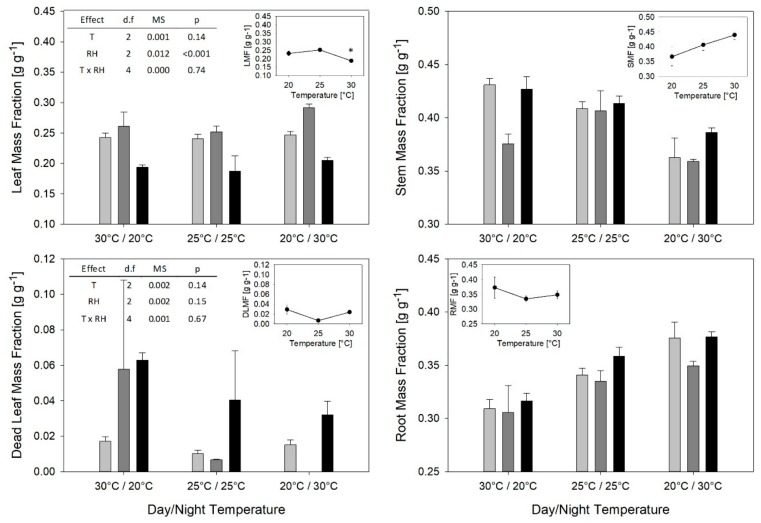
Leaf, stem, dead leaves, and root mass fractions after two weeks of varying day and night temperature and RH regimes. Light grey bars: RH day/night 40/90%; dark grey bars: RH day/night 65/65%; black bars: RH day/night 90/40%. Insert 1: Respective mass fractions at different constant day and night temperatures and constant 65% RH. Asterisk indicates significant difference at *p* < 0.05. Insert 2: Analysis of variance for the respective mass fraction at different day and night temperature and RH regimes. In both, main figure and insert 1, mean values with standard errors (*n* = 3) are presented. Abbreviations: T: temperature; d.f.: degrees of freedom; MS: mean squares.

**Figure 2 plants-08-00521-f002:**
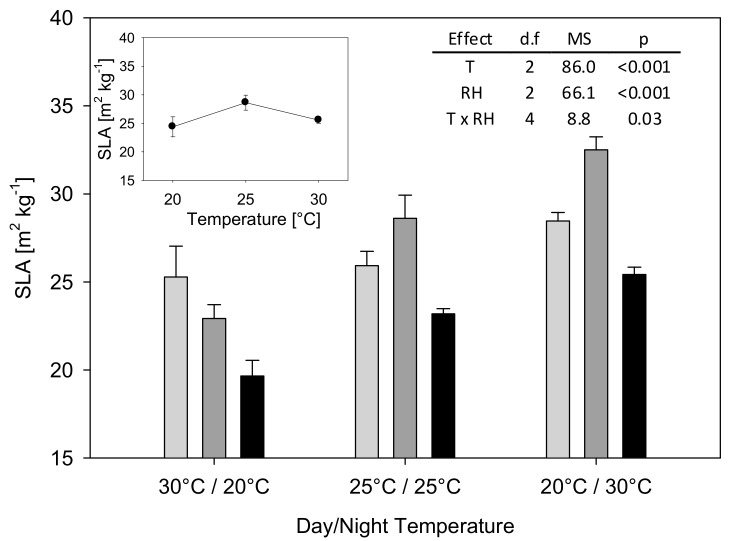
Average plant specific leaf area (SLA) after two weeks exposure to varying day and night temperature and RH regimes. Light grey bars: RH day/night 40/90%; dark grey bars: RH day/night 65/65%; black bars: RH day/night 90/40%. Insert 1: SLA at different constant day and night temperatures and constant 65% RH. Insert 2: Analysis of variance for SLA at different day and night temperatures and RH regimes. In both, main figure and insert 1, mean values with standard errors (*n* = 3) are presented. Abbreviations: T: temperature; d.f.: degrees of freedom; MS: mean squares.

**Figure 3 plants-08-00521-f003:**
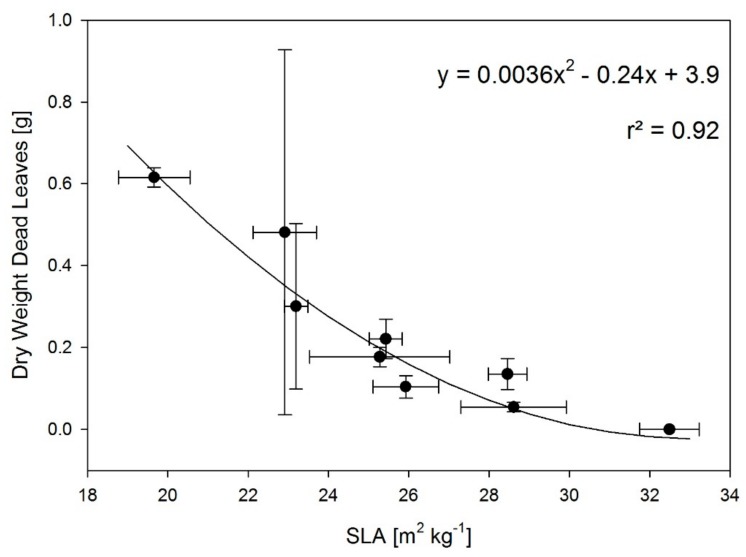
Correlation between SLA and dry weight of dead leaves per plant for plants subjected to different day and night temperatures and RHs. Mean values with standard errors of three replications are presented for 9 treatments, (*n* = 9), i.e., treatments with the same daily mean temperature.

**Figure 4 plants-08-00521-f004:**
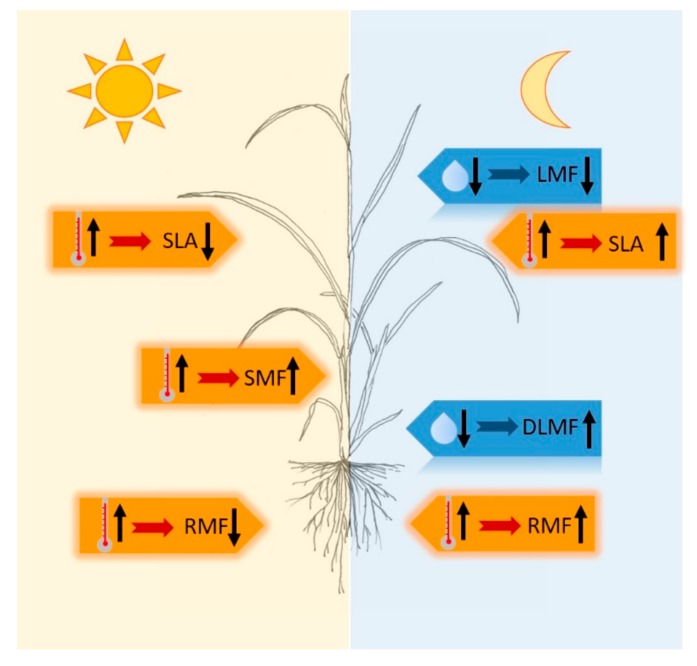
Effects of temperature and relative air humidity during day and night on dry matter partitioning and specific leaf area in rice. Orange pentagons represent temperature effects, blue pentagons represent effects of relative air humidity. Arrow up: high; arrow down: low. Abbreviations: LMF: leaf mass fraction; SMF: stem mass fraction; RMF: root mass fraction; DLMF: fraction of dead leaves; SLA: specific leaf area.

**Table 1 plants-08-00521-t001:** Total plant dry matter and plant leaf area at 49 days after sowing after two weeks exposure to varying day and night temperature and RH regimes. Mean values of three replications are presented including standard errors. Small letters indicate differences at *p* < 0.05 between temperature treatments; capitals indicate differences at *p* < 0.05 between RH treatments.

RH Day/Night [%]	T Day/Night [°C]	Total Dry Matter [g]				Leaf Area [cm^2^]				
40/90	30/20	10.4	±	0.8	A	632	±	14		A
	25/25	10.2	±	0.8		629	±	20		A
	20/30	8.6	±	0.9		601	±	36		B
65/65	30/20	6.0	±	1.4	B	341	±	29	c	B
	25/25	7.9	±	1.3		558	±	39	b	A
	20/30	9.4	±	0.5		890	±	11	a	A
90/40	30/20	9.8	±	0.3	AB	372	±	5		B
	25/25	7.6	±	0.1		333	±	51		B
	20/30	7.0	±	0.5		363	±	22		C

**Table 2 plants-08-00521-t002:** Analysis of variance for plant dry matter and plant leaf area under different day and night temperature and RH regimes around the same daily mean temperature (25 °C) and RH (65%) in 3 replications. Abbreviations: T: temperature; d.f.: degree of freedom; MS: mean square.

Parameter	Effect	d.f	MS	*p*
Dry Matter	T	2	0.35	0.84
	RH	2	9.76	0.02
	T × RH	4	9.01	0.01
Leaf Area	T	2	6.7 × 10^4^	<0.001
	RH	2	1.9 × 10^5^	<0.001
	T × RH	4	8.2 × 10^4^	<0.001

**Table 3 plants-08-00521-t003:** Total plant dry matter and plant leaf area at 49 days after sowing after two weeks exposure to different constant day and night temperatures at 65% RH. Mean values of three replications are presented including standard errors. Small letters indicate differences at *p* < 0.05 between temperature treatments.

RH Day/Night [%]	T Day/Night [°C]	Total Dry Matter [g]				Leaf Area [cm^−2^]			
65/65	20/20	2.9	±	1.4	b	162	±	67	b
	25/25	7.9	±	1.3	a	558	±	39	a
	30/30	9.2	±	0.4	a	439	±	13	a

**Table 4 plants-08-00521-t004:** Pearson correlation coefficients for correlations between day and night growing conditions and SLA and organ mass fractions, respectively, including mean values of all 11 treatments (*n* = 11).

Growing Conditions	SLA	LMF	SMF	RMF	DLMF
T day	−0.57	−0.30	0.76 **	−0.84 **	0.47
RH day	−0.43	−0.55	0.11	0.13	0.59
T night	0.65 *	0.01	−0.20	0.63 *	−0.53
RH night	0.43	0.55	−0.11	−0.13	−0.59

**, *: significant at *p* < 0.01, < 0.05, respectively.

**Table 5 plants-08-00521-t005:** Pearson correlation coefficients for correlations between leaf area (LA) and dry weight (DW) increases per day between 35 and 49 day after sowing and SLA and organ mass fractions, respectively, including mean values all treatments (*n* = 9) with the same daily mean temperature.

Growth Rates	SLA	LMF	SMF	RMF	DLMF
LA day^−1^	0.88 **	0.85 **	−0.37	0.02	−0.86 **
DW day^−1^	0.39	0.39	0.30	−0.24	−0.66

**: significant at *p* < 0.01.

**Table 6 plants-08-00521-t006:** Cultivation conditions for rice plants starting 35 days after sowing until 49 days after sowing.

RH Day/Night [%]	T Day/Night [°C]	Treatment	
40/90	30/20	Tnat	RHnat
	25/25	Tcon	RHnat
	20/30	Tinv	RHnat
65/65	30/20	Tnat	RHcon
	25/25	Tcon	RHcon
	20/30	Tinv	RHcon
	20/20	Tcon-l	RHcon
	30/30	Tcon-h	RHcon
90/40	30/20	Tnat	RHinv
	25/25	Tcon	RHinv
	20/30	Tinv	RHinv

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
