# Peer review of "Responses of Rice Growth to Day and Night Temperature and Relative Air Humidity—Dry Matter, Leaf Area, and Partitioning"

_plants, 2019, doi:10.3390/plants8110521_

Round 1

Reviewer 1 Report

see attached

Reviewer 2 Report

Title:Responses of rice growth to day and night temperature and relative air humidity – Dry matter, leaf area and partitioning

This work focused on analysis of different morphological parameters of rice such as: leaf area, total dry matter, leaf mass, stem mass etc. under different temperatures and relative air humidity regime during the day and night. They observed that the relative air humidity affected plant dry matter and leaf area more than temperature, and under high day temperature, the shoot mass was increased, but the root mass was decreased. In addition, growth rates of rice organs were observed to increase at high night humidities and temperature, thus this study gives greater insight into prediction and study of rice growth under Climate Change.

Minor comments

The different abbreviation used in this article, for example Tnat or RHcon in line 90-92, have no explanation, please give the full description for the abbreviations, the first time they are mention. Why was table 3 separated from table 1, some results were repeated several times (for example, regime 25/25), avoid the repetitions On Table 1 in RH day/night and T day/night section, only the ratios are written e.g. 40/90 or 30/20 etc., please add the units for each characteristic. Additional statistical analysis information as it was not clear how many samples were used for the analysis, please add n (number of variance). In Table 2 it is not written which T and humidity regimes were used for analysis, please add these information in the description of table 2. In Fig 1 showed the Dead Leaf Mass Fraction [g g-1], in material and methods this was not described: were the leaves yellow or completely dry and dead? Missing link to the figures in lines 126-133 and in line 134-140 Most of the analysis was based on morphological parameter, and the discussion is mostly based on the citations of other research work. Giving the additional results, if there are,  based on molecular methods (expression of some marker genes, or molecular components of chlorophyll, the remobilization of nutrients), or physiological methods (such as measurement of ROS, or stress markers, or stomatal closure/conductance) will strengthen this article and bolster the discussion on mechanism of effect of different temperature and humidity regime on rice plant. In the discussion, in line 239-246, all speculation, without citations or data from obtained results. You need to prove all the written statements by citation or your results. Add the description on how and which leaves were measured for area and mass in the materials and methods section. Also, you could build a scheme presenting the effects of humidity and temperature regimes on whole and different plant organs. It could give more attractive value.

Round 2

Reviewer 2 Report

Title: “Responses of rice growth to day and night temperature and relative air humidity – Dry matter, leaf area and partitioning

The different abbreviation used in this article, for example Tnat or RHcon in line 90-92, have no explanation, please give the full description for the abbreviations, the first time they are mentione.

Corrected as requested

Why was table 3 separated from table 1, some results were repeated several times (for example, regime 25/25), avoid the repetitions

Didn’t change

On Table 1 in RH day/night and T day/night section, only the ratios are written e.g. 40/90 or 30/20 etc., please add the units for each characteristic.

Added information as requested

Additional statistical analysis information as it was not clear how many samples were used for the analysis, please add n (number of variance).

Provided information about replication, but didn’t give the information related to the variance

In Table 2 it is not written which T and humidity regimes were used for analysis, please add these information in the description of table 2.

Corrected as requested

In Fig 1 showed the Dead Leaf Mass Fraction [g g-1], in material and methods this was not described; were the leaves yellow or completely dry and dead?

Added the suggested description

Missing link to the figures in lines 126-133 and in line 134-140

Added the missing links for figures

Most of the analysis was based on morphological parameter, and the discussion is mostly based on the citations of other research work. Giving the additional results, if there are,  based on molecular methods (expression of some marker genes, or molecular components of chlorophyll, the remobilization of nutrients), or physiological methods (such as measurement of ROS, or stress markers, or stomatal closure/conductance) will strengthen this article and bolster the discussion on mechanism of effect of different temperature and humidity regime on rice plant.

Didn’t add any requested information

In the discussion, in line 239-246, all speculation, without citations or data from obtained results. You need to prove all the written statements by citation or your results.

Corrected

Add the description on how and which leaves were measured for area and mass in the materials and methods section.

Corrected as requested

Also, you could build a scheme presenting the effects of humidity and temperature regimes on whole and different plant organs. It could give more attractive value.

Added the suggested scheme